# Crystal Structure of Human CD47 in Complex with Engineered SIRPα.D1(N80A)

**DOI:** 10.3390/molecules27175574

**Published:** 2022-08-30

**Authors:** Jifeng Yu, Song Li, Dianze Chen, Dandan Liu, Huiqin Guo, Chunmei Yang, Wei Zhang, Li Zhang, Gui Zhao, Xiaoping Tu, Liang Peng, Sijin Liu, Xing Bai, Yongping Song, Zhongxing Jiang, Ruliang Zhang, Wenzhi Tian

**Affiliations:** 1Department of Hematology, The First Affiliated Hospital of Zhengzhou University, Zhengzhou 450052, China; 2ImmuneOnco Biopharmaceuticals (Shanghai) Co., Ltd., Shanghai 201203, China

**Keywords:** cancer immunotherapy, SIRPα-Fc fusion proteins, CD47/SIRPα, crystal structure, immune checkpoint pathway, computer-aided drug discovery

## Abstract

**Background:** Targeting the CD47/SIRPα signaling pathway represents a novel approach to enhance anti-tumor immunity. However, the crystal structure of the CD47/SIRPα has not been fully studied. This study aims to analyze the structure interface of the complex of CD47 and IMM01, a novel recombinant SIRPα-Fc fusion protein. **Methods:** IMM01-Fab/CD47 complex was crystalized, and diffraction images were collected. The complex structure was determined by molecular replacement using the program PHASER with the CD47-SIRPαv2 structure (PDB code 2JJT) as a search model. The model was manually built using the COOT program and refined using TLS parameters in REFMAC from the CCP4 program suite. **Results:** Crystallization and structure determination analysis of the interface of IMM01/CD47 structure demonstrated CD47 surface buried by IMM01. Comparison with the literature structure (PDB ID 2JJT) showed that the interactions of IMM01/CD47 structure are the same. All the hydrogen bonds that appear in the literature structure are also present in the IMM01/CD47 structure. These common hydrogen bonds are stable under different crystal packing styles, suggesting that these hydrogen bonds are important for protein binding. In the structure of human CD47 in complex with human SIRPα, except SER66, the amino acids that form hydrogen bonds are all conserved. Furthermore, comparing with the structure of PDB ID 2JJT, the salt bridge interaction from IMM01/CD47 structure are very similar, except the salt bridge bond between LYS53 in IMM01 and GLU106 in CD47, which only occurs between the B and D chains. However, as the side chain conformation of LYS53 in chain A is slightly different, the salt bridge bond is absent between the A and C chains. At this site between chain A and chain C, there are a salt bridge bond between LYS53 (A) and GLU104 (C) and a salt bridge bond between HIS56 (A) and GLU106 (C) instead. According to the sequence alignment results of SIRPα, SIRPβ and SIRPγ in the literature of PDB ID 2JJT, except ASP100, the amino acids that form common salt bridge bonds are all conserved. **Conclusion:** Our data demonstrated crystal structure of the IMM01/CD47 complex and provides a structural basis for the structural binding interface and future clinical applications.

## 1. Introduction

The cluster of differentiation 47 (CD47) protein, expressed on both healthy and cancer cells, plays a pivotal role in this balance by delivering a “do not eat me signal” upon binding to the signal-regulatory protein alpha (SIRPα) receptor on myeloid cells. CD47 is overexpressed on many types of tumors and acts as an important tumor antigen for the development and progression of various tumors. As a cell surface glycoprotein molecule, CD47 belongs to the immunoglobulin superfamily and binds to various proteins, including integrin, thrombospondin-1 and SIRPα. Thus, overexpression of CD47 enables tumor cells to evade immune surveillance via the blockade of phagocytic mechanisms [1]. Meanwhile, macrophages play an important role in the immune system by phagocytosis mediated by various pathways such as phosphatidylserine (PS) extracellular exposure, representing an “eat me” signal for macrophages to activate the phagocytosis [2]. The CD47–SIRPα axis regulates homeostatic processes by controlling myeloid cell-mediated removal of aging cells, erythrocytes, hematopoietic stem cells and neuronal synapses [3]. SIRPα sends signals that inhibit the prophagocytic, “eat me” signals transmitted by (i) phosphatidylserine, (ii) the engagement of antibody–antigen complexes with the FcR and (iii) the interaction of calreticulin expressed on tumor cells with the LDL receptor-related protein (LRP) receptor. Blockade of the CD47/SIRPα interaction can inhibit tumor growth by inhibiting the suppressive effects of SIRPα and promoting macrophage activity.

The SIRP family of proteins consists of five members: SIRPα, SIRPβ1, SIRPβ2, SIRPγ and SIRPδ [3]. SIRPα (also known as PTPNS1, SHPS1, CD172A and P84), which can bind to CD47, contains an extracellular region with three immunoglobulin superfamily (IgSF) domains, including an NH2-terminal ligand-binding V-domain [3]. Allelic variants with polymorphisms in the ligand-binding domain have been reported in the African, Japanese, Chinese and Caucasian populations, three of which (SIRPαV1, SIRPαV2 and SIRPαV8) are the most prominent haplotypes among the human population, jointly covering approximately 90% [4,5]. The binding of SIRP with its ligand CD47 regulates leukocyte functions, including transmigration, phagocytosis and cytokine secretion. Recent progress has provided significant insights into the structural details of the distal IgV domain (D1) of SIRP [6]. 

A few safety concerns regarding the RBC binding issue from anti-CD47 antibodies have been raised, which include hematogglutination, antibody-dependent cell-mediated cytotoxicity (ADCC)/antibody-dependent cellular phagocytosis (ADCP) (if IgG1-Fc) against red blood cells and T cell apoptosis upon binding with CD47 antibody [7]. SIRPα-Fc fusion proteins have been considered as having more therapeutic potential and less adverse effects than anti-CD47 antibodies. Among the 10 allelic variants of SIRPα, SIRPα V1 and V2 are the most prevalent variants [8]. To date, six SIRPα fusion proteins including ours are currently in phase I or phase II clinical trials [9]. Among these SIRPα fusion proteins, IMM01 has a V2D1 variant formation with an N80A mutation [1,9]. Regarding the crystal structures of the CD47/SIRPα complex, Hatherley et al. described the high-resolution X-ray crystallographic structures of the immunoglobulin superfamily domain of CD47 alone and in complex with the N-terminal ligand-binding domain of SIRPα. This explained the specificity of CD47 for the SIRP family of paired receptors in atomic detail [10]. However, few studies have been carried out regarding the crystal structure of the CD47/SIRPα-Fc fusion protein complex. In this study, we explore the crystal structure of CD47/IMM01, a novel SIRPα-Fc fusion protein.

## 2. Results

### 2.1. SDS-PAGE Analysis Results of CD47-His 

SDS-PAGE analysis of CD47-His showed a molecular weight of ~34 kDa after purification. The theoretical molecular weight of CD47-His is 14.8 kDa. The SDS-PAGE results show that the actual molecular weight is ~34 kDa due to the severe glycosylation modification of CD47. Based on the result, fractions of flow through 2–14 were pooled together and concentrated by an ultrafiltration tube (Ultracel-10 regenerated cellulose membrane, Merck, Cat#UFC801008) with a molecular cut-off value of 10 KDa (Figure 1).

### 2.2. SDS-PAGE Analysis Results of Purified IMM01-Fab 

Number of amino acids in IMM01 was 730 with molecular weight of 80561.17 and theoretical pI of 7.35 (Figure 2). Sequence of IMM01 is as follows:

EEELQVIQPDKSVSVAAGESAILHCTVTSLIPVGPIQWFRGAGPARELIYNQKEGHFPRVTTVSESTKRENMDFSISISAITPADAGTYYCVKFRKGSPDTEFKSGAGTELSVRAKPSAPVVSGPAARATPQHEPKSCDKTHTCPPCPAPELLGGPSVFLFPPKPKDTLMISRTPEVTCVVVDVSHEDPEVKFNWYVDGVEVHNAKTKPREEQYNSTYRVVSVLTVLHQDWLNGKEYKCKVSNKALPAPIEKTISKAKGQPREPQVYTLPPSRDELTKNQVSLTCLVKGFYPSDIAVEWESNGQPENNYKTTPPVLDSDGSFFLYSKLTVDKSRWQQGNVFSCSVMHEALHNHYTQKSLSLSPGK.

After Papain digestion, the number of amino acids in the IMM01-Fab is 142 with a molecular weight of 15,241.15, theoretical pI of 7.15 and extinction coefficient of 0.662. The sequence is as follows:

EEELQVIQPDKSVSVAAGESAILHCTVTSLIPVGPIQWFRGAGPARELIYNQKEGHFPRVTTVSESTKRENMDFSISISAITPADAGTYYCVKFRKGSPDTEFKSGAGTELSVRAKPSAPVVSGPAARATPQHEPKSCDKTH.

### 2.3. Confirmation of IMM01 Fusion Protein Expression and Its Molecular Weight

With the special molecular design of IMM01, recombinant SIRPα-Fc fusion protein was made through the CHO-K1 cell expression system. After expression of fusion protein, IMM01 was isolated by using an affinity purification process. The SDS-PAGE method was used to analyze the size of the IMM01 and IMM01/CD47 complex. SE-HPLC was used to evaluate the purity of fusion proteins. A size-exclusion chromatography (SEC) method was used for IMM01-Fab/CD47 complex purification. A Superdex200 column was used to increase purity of the IMM01-Fab/CD47 complex. Fractions 23–26 were concentrated for crystallization (Figure 3).

### 2.4. Crystallization and Structure Determination

Crystal observation was carried out in two different conditions: IMM01-Fab/CD47his-CSHT-B5 and IMM01-Fab/CD47his-PR-G4 (Figure 4). The 2.76 Å data were used for the analysis of structure determination. The crystal mounting condition and diffraction test criteria are listed in the Appendix A.

### 2.5. IMM01/CD47 Complex Interface Analysis

Well-diffracting crystals were obtained for the complex of CD47 and IMM01. Details of the common hydrogen bonds between IMM01 and CD47 are given in Table 1. The salt bridge bonds between IMM01-Fab and CD47 are given in Table 2. Furthermore, the water-mediated hydrogen bond between IMM01 and CD47 is given in Table 3. The crystal structures of the CD47/IMM01 complex for the hydrogen bond interactions between chain A (IMM01-Fab) and chain C (CD47) are shown in Figure 5. The salt bridge bond interactions between chain A (IMM01) and chain C (CD47) are shown in Figure 6. Further analysis of the CD47 binding interface of IMM01/CD47 structure demonstrated a CD47 surface buried by IMM01 (Figure 7, Figure 8, Figure 9). IMM01/CD47 complex interface analysis showed that compared with a previous publication (PDB ID 2JJT), the interactions of IMM01/CD47 structure are the same. All the hydrogen bonds that appear in the structure in the literature are also present in the IMM01/CD47 structure. These common hydrogen bonds are stable under different crystal packing styles, suggesting that these hydrogen bonds are important for protein binding. According to the sequence alignment results of SIRPα, SIRPβ and SIRPγ in the previous publication (PDB ID 2JJT), except SER66, the amino acids that form hydrogen bonds are all conserved. Furthermore, compared with the structure in the literature (2JJT), the salt bridge interactions of the IMM01/CD47 structure are very similar, except the salt bridge bond between LYS53 in IMM01 and GLU106 in CD47, which only occurs between the B and D chains. However, as the side chain conformation of LYS53 in chain A is slightly different, the salt bridge bond is absent between the A and C chains. At this site between chain A and chain C, there are a salt bridge bond between LYS53 (A) and GLU104 (C) and a salt bridge bond between HIS56 (A) and GLU106 (C) instead. According to the sequence alignment results of SIRPα, SIRPβ and SIRPγ in the previous publication (PDB ID 2JJT), except ASP100, the amino acids that form common salt bridge bonds are all conserved. The interface area between chain A (IMM01) and chain C (CD47) is 973.5Å2, which is composed of 29 residues from IMM01 and 27 residues from CD47, including 14 hydrogen bonds (Table 1) and 11 salt bridges (Table 2). The interface area between chain B (IMM01) and chain D (CD47) is 965.8 Å2, which is composed of 27 residues from IMM01 and 26 residues from CD47, including 14 hydrogen bonds (Table 1) and 10 salt bridges (Table 2). Interface analysis was performed using the qtPISA program in the CCP4 suite.

Table 1: Comparing IMM01/CD47 structure with the structure in the literature (PDB ID 2JJT), the interactions are the same, and all the hydrogen bonds that appear in the structure in the literature are also present in this structure. These common hydrogen bonds are chain A:C and chain B:D. These hydrogen bonds are stable under different crystal packing styles, suggesting that these hydrogen bonds are important for protein binding. According to the sequence alignment results of SIRPα, SIRPβ and SIRPγ in the literature (PDB ID 2JJT), except SER66, the amino acids that form hydrogen bonds are all conserved. 

Table 2: Comparing IMM01/CD47 structure with the structure in the literature (PDB ID 2JJT), the salt bridge interactions are very similar, except the salt bridge bond between LYS53 in IMM01 and GLU106 in CD47, which only occurs between the B and D chains. However, as the side chain conformation of LYS53 in chain A is slightly different, the salt bridge bond is absent between the A and C chains. At this site between chain A and chain C, there are a salt bridge bond between LYS53 (A) and GLU104 (C) and a salt bridge bond between HIS56 (A) and GLU106 (C) instead. These common salt bridges are chain A:C and chain B:D. According to the sequence alignment results of SIRPα, SIRPβ and SIRPγ in the literature (PDB ID 2JJT), except ASP100, the amino acids that form salt bridge bonds are all conserved.

Table 3: Comparing IMM01/CD47 structure with the structure in the literature (PDB ID 2JJT), most of water-mediated hydrogen bond interactions are very similar. These common salt bridges are chain A:C and chain B:D.

## 3. Discussion

The high-resolution crystal structure of the CD47 binding to SIRPα through the ligand-binding domain d1 as a complex was first published by Hatherley et al. in 2008. In the Protein Data Bank, 2JJT was designated as the identifier for the CD47/SIRPα complex (PDB ID: 2JJT). The crystal structure results showed that CD47 and SIRPα d1 formed a 1:1 stoichiometry complex [10]. It was confirmed that CD47 and SIRPα d1 molecules are interdigitated with each other so their interaction is mainly mediated by loops at the intracellular side and the structure of the SIRPα inhibitory receptor reveals a binding face reminiscent of that used by T cell receptors [11]. The interaction interface of the CD47/SIRPα d1 complex is mainly formed of four N-terminal loops of the SIRPα d1 domain. The FG loop of CD47 is embedded into the cavity on the surface of SIRPα d1 via Thr102 of the FG loop inserted deep into SIRPα d1 [6]. In addition, the high-affinity SIRPα variant FD6/CD47 complex has been reported [12]. Research showed that the hot spot residue-mediated polar interactions on CD47 comprise Glu97, Thr99, Glu100, Arg103, Glu104 and Glu 106. Among them, Glu104 and Glu106 of CD47 form hydrogen bonds with SIRPα [6]. Furthermore, four complex crystal structures of CD47 monoclonal antibodies (mAbs) have been reported, including the CD47/magrolimab complex (PDB ID:5IWL) [13], CD47/B6H12.2 complex (PDB ID: 5TZU) [14], CD47/C47B222 complex (PDB ID: 5TZ2) [14] and CD47/C47B161 complex (PDB ID: 5TZT) [14]. Magrolimab mainly binds to N-terminal pyroglutamate of CD47 which is critical for CD47/SIRPα interaction and magrolimab binds to the BC and FG loops, which are highly overlapping epitopes with SIRPα [15,16]. Binding to CD47, the SIRPα-Fc fusion protein can block the immunosuppressive CD47–SIRPα signal between macrophages and tumor cells as a decoy receptor and has demonstrated its immunotherapeutic efficacy in various tumors [1]. A study revealed that BC, CC0 and FG loops on CD47 can be the potential binding areas for the potential SIRPα inhibitor therapeutics, and Tyr37, Asp46, Glu97, Glu100 and Glu106 may be developed into binding sites for inhibitors targeting CD47. Among these loops, Glu54, Gly55, Ser66 and Ser98 residues may play crucial roles in the interaction and may act as binding sites for designing future SIRPα inhibitors [6].

CD47 has become a potential therapeutic target and is being studied in various preclinical studies and clinical trials [17,18,19,20]. Targeting CD47/SIRPα represents a novel approach to enhance anti-tumor immunity by augmenting or reactivating critical tumor clearance mechanisms, as a key immune checkpoint in different cancers similar to that of the PD-1/PD-L1 checkpoint for solid tumors. Recently, several anti-CD47 antibodies have been developed and tested in different preclinical and clinical studies. However, clinical application was limited because of the potential hematologic toxicity [1]. Therefore, it is crucial to identify the best binding positions for small molecules, and drug design based on the structural data will lead to the successful development of CD47/SIRPα inhibitors [6]. Recently, CD47/SIRPα inhibitors have aroused enormous interest among researchers. This field is progressing rapidly, and some mAbs targeting the CD47/SIRPα pathway have reached clinical phase II and phase III [21]. Notably, despite the excellent clinical performance shown by CD47/SIRPα antibodies, the limitations of antibody drugs, including poor tumor permeability, undesirable oral bioavailability and poor stability, hinder their clinical application [22,23,24,25]. Targeting the CD47/SIRPα signaling pathway with a different strategy than anti-CD47 mAbs and potentially eliminating the problems caused by antibody drugs, SIRPα-Fc fusion proteins have attracted the attention of researchers and have become a promising research area. According to the literature and ClinicalTrials.gov database, six SIRPα fusion proteins are currently in phase I or phase II clinical trials [9]. Furthermore, efforts have been made to modify the SIRPα-Fc fusion with collagen binding domain (CBD) conjugation. This CBD–SIRPα-Fc conjugate may have the potential to be an effective tumor immunotherapy with improved anti-tumor efficacy but fewer non-tumor-targeted side effects [26]. In vivo distribution experiments showed that CBD–SIRPα-Fc accumulated in tumor tissue more effectively compared to unmodified SIRPα-Fc, probably due to the exposed collagen in the tumor vascular endothelium and stroma resulting from the abnormal vessel structure [26]. A preclinical study regarding a novel oncolytic adenovirus carrying a SIRPα–IgG1 Fc fusion gene (termed SG635-SF) showed that SIRPα-Fc fusion protein enhances the anti-tumor effect of oncolytic adenovirus against ovarian cancer [27]. In addition, another study demonstrated that the SIRPα–αCD123 fusion antibodies targeting CD123 in conjunction with CD47 blockade enhance the clearance of AML-initiating cells [28]. All these results confirmed that targeting the CD47/SIRPα axis can provide a new potential treatment for various cancers.

As a new SIRPα-Fc fusion protein targeting the CD47/SIRPα pathway, IMM01 exhibited strong dual-functional anti-tumor activity through phagocytosis by blocking the “do not eat me” signal and activating the “eat me” signal. IMM01 can be used as a monotherapy or in combination with other targeted immune checkpoint inhibitors. More importantly, removal of N-glycosylation modification improves the consistence and purity of the protein expression, the potential druggability and the potential mitigation of immunogenicity (manuscript is under review for publication). Our data demonstrated the crystal structures of the complex of IMM01 and CD47 on the common hydrogen bonds, the salt bridge interaction and the water-mediated hydrogen bonds. In addition, through the comparison, the consistency between our data and the data published in the literature (PDB ID 2JJT) confirms the IMM01/CD47 structural bonding interface and further provides a structural basis. The clinical application of IMM01 in the treatment of various tumors deserves further exploration.

## 4. Materials and Methods 

### 4.1. Construction, Expression and Purification of CD47-His

#### 4.1.1. Construction of Plasmid for CD47-His Protein Expression

The hCD47(19-135)-His was construct into a pTT5 vector. The Cys at residue 15 was mutated to Gly to minimize disulfide crosslinking. The plasmid was extracted and transiently transfected Expi293F cells to express the target protein [9]. The pTT5-hCD47(19-135)-His AA sequence is as follows. Theoretical isoelectric points (pI)/molecular weight are 6.16/14789.64Da, and the extinction coefficient is 1.155.

MGSTAILGLLLAVLQGGRAQLLFNKTKSVEFTFGNDTVVIPCFVTNMEAQNTTEVYVKWKFKGRDIYTFDGALNKSTVPTDFSSAKIEVSQLLKGDASLKMDKSDAVSHTGNYTCEVTELTREGETIIELKYRVVSTRHHHHHH*

#### 4.1.2. Protein Expression of CD47-His (for 1 L)

(1)Twenty-four hours prior to transfection, split Expi293F cells back to a density of 2 × 10^6^ cells/mL. Ensure cells are in healthy, log phase growth and viability is >95%.(2)On the day of transfection, split cells to a density of 2.7–3.0 × 10^6^ cells/mL.(3)Using the general rule, 1 μg DNA is used per 1 mL of culture, 1 mg of DNA is used for transfection of 1 L culture.(4)Dilute 1 mg of pTT5-hCD47(19-135)-His plasmid into a final volume of 50 mL Opti-MEM medium.(5)Dilute 6 mg of PEI transfection reagent into a final volume of 50 mL Opti-MEM medium. Incubate at room temperature for 3 min; the solution will become slightly cloudy.(6)Mix 10 mL of DNA with 10 mL of PEI and incubate for 15 min at room temperature.(7)Add 100 mL of the DNA:PEI complex dropwise to the culture. This will bring the final density of the culture to ~2.5 × 10^6^ cells/mL.(8)Incubate on an orbital shaking platform at 37 °C with 8% CO_2_ at a speed of 125 rpm.(9)After incubating cells for 20 h, add cell transfection enhancer to the culture.(10)Harvest cell supernatants at 120 h post-transfection.

#### 4.1.3. Purification of CD47-His

The expressed supernatant was clarified by centrifugation. The resin was pre-equilibrated with buffer A (1xPBS pH 7.4, KH_2_PO_4_ 2 mM; Na_2_HPO_4_ 8 mM; NaCl 136 mM; KCL 2.6 mM), then the supernatant flowed through the resin overnight at 4 °C. The column was washed with buffer A, buffer A + 20 mM imidazole, buffer A + 300 mM imidazole, proteins were detected by G-250, 12% SDS-PAGE was run for analysis. 

IMM01-Fab was digested with papain to isolate Fc fragments.

#### 4.1.4. Q HP Purification

About 8 mL of the elute was concentrated, and diluted with about 60 mL of buffer B (50 mM Tris, 10% glycerol, pH 8.0), then supernatant was loaded onto a pre-equilibrated 1 mL HiTrap Q HP column (General Electric Co., Boston, MA, US), flow through was collected, followed by gradient elution by combining buffer B and buffer C (50 mM Tris, 500 mM NaCl, pH 8.0)

### 4.2. Construction and Expression of Recombinant Protein IMM01

IMM01 is the recombinant fusion protein of SIRPα (V2) extracellular segment domain 1 and human IgG1 Fc, with the N-glycosylation site N80 mutated to alanine (A) in the D1 region. The nucleotide sequence encoding each fusion protein was synthesized by GenScript and was subcloned into a mammalian expression vector. IMM01 fusion proteins were produced separately using a CHO-K1 cell expression system developed in house (ATCC# CCL-61). Fusion proteins were purified with protein A affinity column chromatography. The purity of these fusion proteins was evaluated by size-exclusion high-performance liquid chromatography (SE-HPLC). A function study demonstrated that the novel SIRPα-Fc fusion protein IMM01 exhibits dual anti-tumor activities by blocking “do not eat me” signals and activating the “eat me” signals to enhance the phagocytosis by targeting the CD47/SIRPα signal pathway. The detailed information regarding the IMM01 characterization and functions has been summarized in another manuscript which is under review for publication.

IMM01-Fab was purified by the protein A affinity purification method. A Superdex200 column was used to increase purity of IMM01-Fab. For purification of the IMM01-Fab/CD47 complex, IMM01-Fab and CD47-His were mixed in molar ratio of 1:1 and incubated at 4 °C overnight. Then, a size-exclusion chromatography (SEC) method was used for IMM01-Fab/CD47 complex purification. A Superdex200 column was used to increase purity of the IMM01-Fab/CD47 complex.

### 4.3. Crystallization of IMM01-Fab/CD47 Complex, Data Collection, Structure Determination and Refinement

(a)The hCD47(19-135)-His was constructed into a vector and expressed in Expi293F cells. The Cys at residue 15 was mutated to Gly to minimize disulfide crosslinking. Recombinant CD47 proteins were purified by nickel affinity chromatography and ion exchange chromatography.(b)IMM01-Fab was fused with the Fc tag in a C-terminal construct into a vector and expressed in CHO cells. The proteins were purified by protein A affinity chromatography and digested with papain to remove the Fc tag.(c)IMM01-Fab and CD47 were mixed in a 1:1 molar ratio, deglycosylated using endoglycosidase Hf and concentrated to contain each protein at 12.6 mg/mL. Sitting drop vapor diffusion crystallization experiments were performed using an NT8 robot to dispense nanoscale protein precipitant drops that were equilibrated against precipitant reservoirs at 18 °C. Crystals were cryoprotected in mother liquor supplemented with 20% glycerol and flash-frozen in liquid nitrogen.(d)Diffraction images were collected at 100 K at Beamline 19-ID-D of the Advanced Photon Source (APS, USA) using a Pilatus3 × 6M detector. The dataset, at the resolution of 2.76 Å, was processed with the XDS programs [29] and scaled with AIMLESS from the CCP4 suite [30]. The unit cell parameters for the crystal were a =53.58 Å, b = 99.64 Å, c = 138.93 Å, α = β = γ = 90.0°, in space group *P*2_1_2_1_2_1_ with two copies of the molecules in the IMM01-Fab/CD47 complex in one asymmetric unit. The complex structure was determined by molecular replacement using the program PHASER with the CD47-SIRPαv2 structure (Protein Data Bank identification, PDB ID 2JJT) as a search model. The model was manually built using the COOT program [31] and refined using TLS parameters in REFMAC from the CCP4 program suite [32]. All structural figures were prepared with PyMol [33].

## 5. Conclusions

Using a recombinant technology, we were able to make a special recombinant SIRPα-Fc fusion protein named IMM01 with a special SIRPα-Fc structure, which has SIRPα (V2) extracellular segment domain 1 and human IgG1 Fc, with the N-glycosylation modification site of N80A mutation in the D1 region. The structure interface analysis of the IMM01/CD47 complex shows the common hydrogen bonds are stable under different crystal packing styles, suggesting that these hydrogen bonds are important for protein binding. The salt bridge interaction of the IMM01/CD47 structure is similar to the structure in the literature of PDB ID 2JJT, the structure of human CD47 in complex with human SIRPα. Separate results to be published show that IMM01 has strong dual-functional anti-tumor activity through phagocytosis and removal of N-glycosylation modification improves the potency. This provides a structural basis for the IMM01/CD47 structural binding interface and provides a scientific theoretical basis for IMM01’s clinical application in the treatment of various tumors.

## Figures and Tables

**Figure 1 molecules-27-05574-f001:**
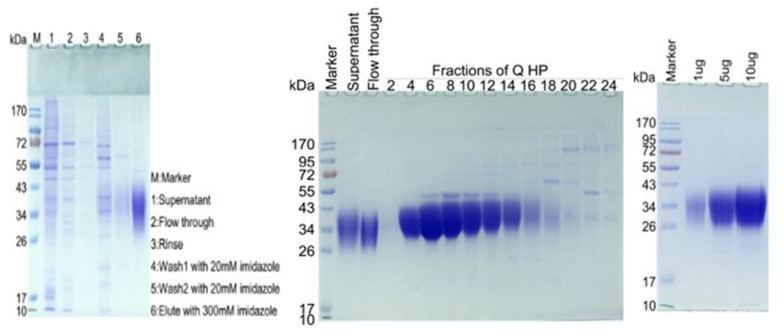
SDS-PAGE analysis of CD47-His after the Ni column showed a molecular weight of ~34 kDa (**left** figure). Based on the result, fractions of flow through 4–14 were pooled together and concentrated by an ultrafiltration tube with a molecular cut-off value of 10 kDa (**middle** figure). The SDS-PAGE analysis of final CD47-His showed a clean band with a molecular weight of ~34 kDa (**right** figure). The theoretical molecular weight of CD47-His is 14.8 kDa. The SDS-PAGE re-sults show that the actual molecular weight is ~34 kDa due to the severe glycosylation modifica-tion of CD47.

**Figure 2 molecules-27-05574-f002:**
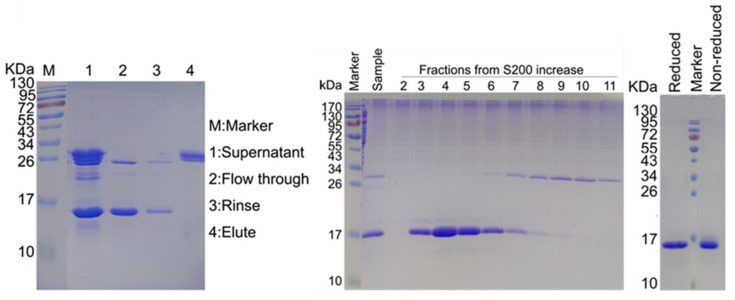
Protein A purification method was used to purify the IMM01-Fab from the supernatant (**left** figure). Superdex200 column was used increase purity of IMM01-Fab. Fractions 3, 4, 5 were pooled together for concentration (**middle** figure). SDS-PAGE analysis of final purified IMM01-Fab showed a clean single band (**right** figure).

**Figure 3 molecules-27-05574-f003:**
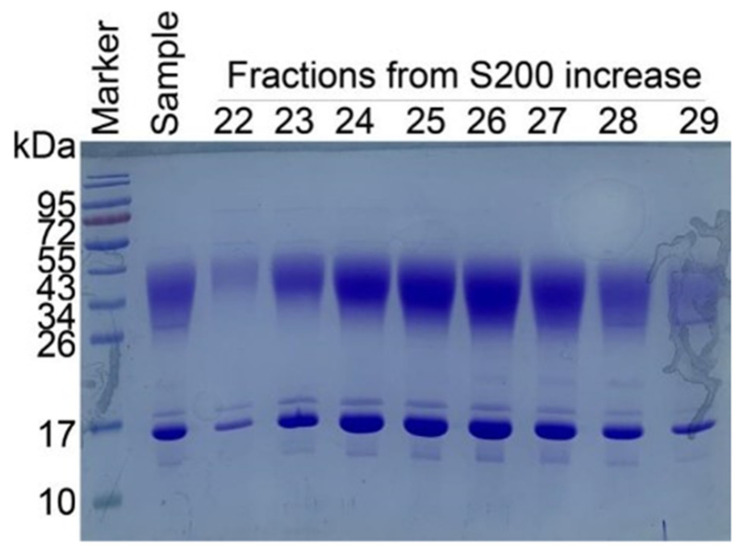
IMM01-Fab and CD47-His were mixed in molar ratio 1:1 and incubated at 4 °C overnight. The size-exclusion chromatography (SEC) method was used for IMM01-Fab/CD47 complex purification. Superdex200 column was used to increase purity of IMM01-Fab/CD47 complex. Fractions 23–26 were concentrated for crystallization.

**Figure 4 molecules-27-05574-f004:**
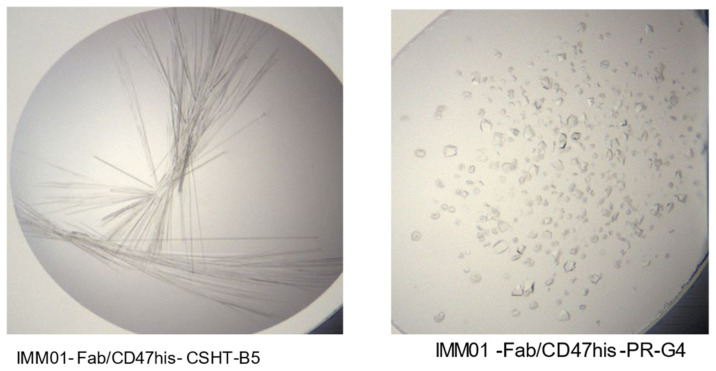
Crystal observation in two different conditions, IMM01-Fab/CD47his-CSHT-B5 (**left** figure) and IMM01-Fab/CD47his-PR-G4 (**right** figure).

**Figure 5 molecules-27-05574-f005:**
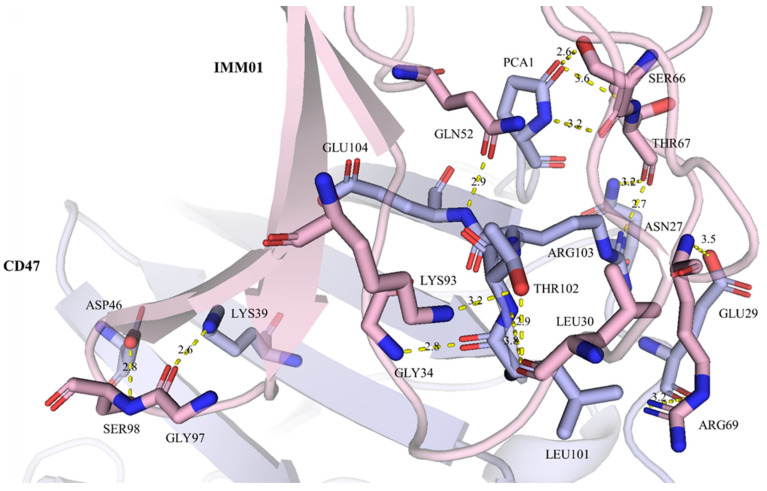
Hydrogen bond interactions between chain A (IMM01) and chain C (CD47). Chain A and chain C are shown in pink and light blue, respectively. The residues forming hydrogen bonds are shown as sticks, key hydrogen bonds are highlighted as yellow dashed lines.

**Figure 6 molecules-27-05574-f006:**
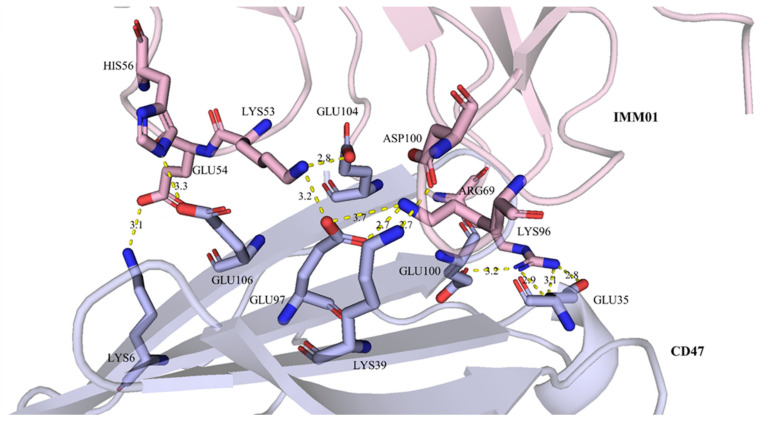
Salt bridge bond interactions between chain A (IMM01) and chain C (CD47). Chain A and chain C are shown in pink and light blue, respectively. The residues forming salt bridge bonds are shown as sticks, salt bridge bonds are highlighted as yellow dashed lines.

**Figure 7 molecules-27-05574-f007:**
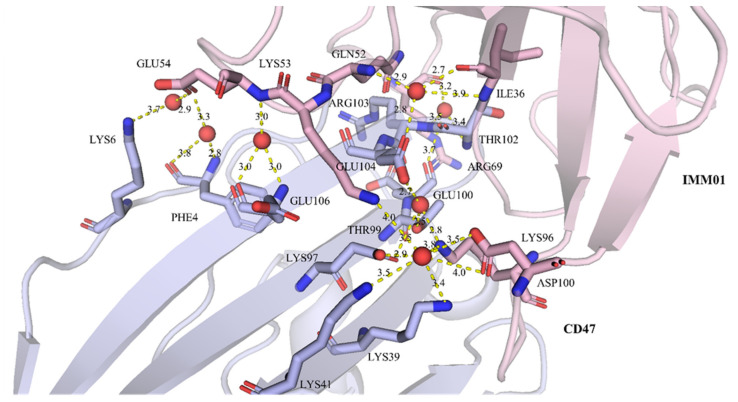
Water-bridged hydrogen bond interactions between chain A (IMM01) and chain C (CD47). Chain A and chain C are shown in pink and light blue, respectively. The residues forming hydrogen bond with waters are shown as sticks, and the water molecules are shown as red spheres. Key hydrogen bonds are highlighted as yellow dashed lines.

**Figure 8 molecules-27-05574-f008:**
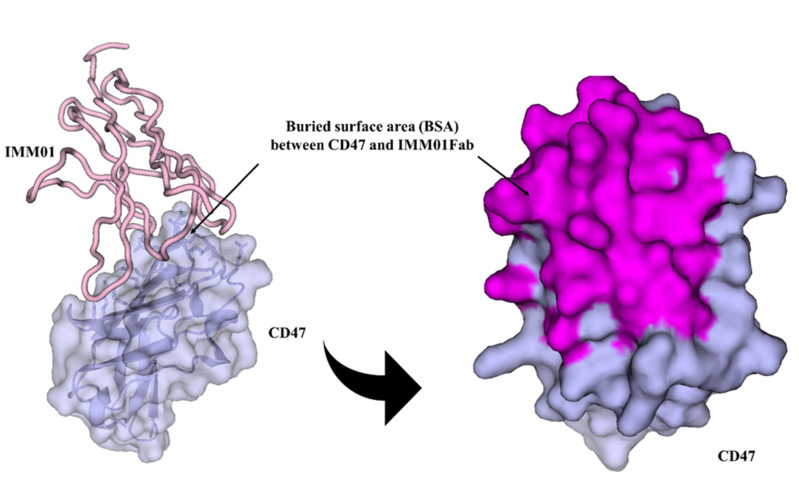
Buried surface area (BSA) between chain A (IMM01) and chain C (CD47). Chain A and chain C are shown in pink and light blue, respectively. The CD47 is shown as surface, and the interfacing residues are highlighted in magenta.

**Figure 9 molecules-27-05574-f009:**
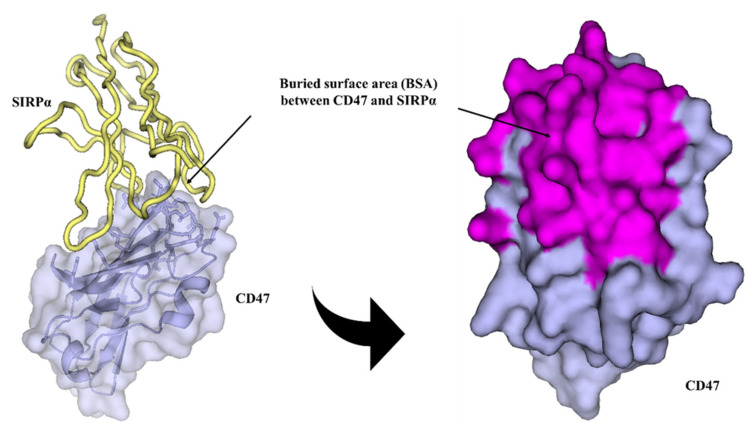
Buried surface area (BSA) between chain A (SIRPα, 2JJS) and chain C (CD47, 2JJS). Chain A and chain C are shown in yellow and light blue, respectively. The CD47 is shown as surface, and the interfacing residues are highlighted in magenta.

**Table 1 molecules-27-05574-t001:** Common hydrogen bonds between IMM01 and CD47.

IMM01		CD47		Distance (Å)
Residue	Atom	Residue	Atom	
A:SER66	O	C:GLN1	N	2.7
A:SER66	OG	C:GLN1	NE2	2.4
A:THR67	O	C:ASN27	ND2	3.1
A:GLY97	O	C:LYS39	NZ	2.6
A:LEU30	O	C:THR102	OG1	2.9
A:THR67	O	C:ARG103	NH1	2.6
A:GLN52	OE1	C:GLU104	N	2.8
A:GLY34	N	C:LEU101	O	2.7
A:ARG69	N	C:GLU29	OE2	3.4
A:ARG69	NH1	C:GLU29	O	3.3
A:SER98	OG	C:ASP46	OD2	2.8
A:LYS93	NZ	C:THR102	OG1	3.2
A:ASP73	OD1	C:TYR113	OH	2.4
A:GLU65	N	C:VAL116	O	3.2
A:SER64	OG	C:VAL116	O	2.7
B:SER66	O	D:GLN1	N	3.3
B:SER66	OG	D:GLN1	NE2	2.2
B:TYR50	OH	D:GLN1	NE2	3.4
B:THR67	O	D:ASN27	ND2	3.3
B:GLY97	O	D:LYS39	NZ	2.7
B:LEU30	O	D:THR102	N	3.8
B:LEU30	O	D:THR102	OG1	2.8
B:THR67	O	D:ARG103	NH1	2.5
B:GLN52	OE1	D:GLU104	N	3.2
B:GLY34	N	D:LEU101	O	2.7
B:GLN52	OE1	D:GLU104	N	3.2
B:ARG69	N	D:GLU29	OE2	3.5
B:SER98	OG	D:ASP46	OD2	2.9
B:LYS93	NZ	D:THR102	OG1	3
B:GLU70	OE2	D:TYR113	OH	2.7
B:GLU70	OE1	D:VAL116	N	3.5
B:ARG69	NH1	D:VAL116	O	2.8

**Table 2 molecules-27-05574-t002:** Salt bridge bonds between IMM01 and CD47.

IMM01		CD47		Distance (Å)
Residue	Atom	Residue	Atom	
A:GLU54	OE2	C:LYS6	NZ	3
A:ASP100	OD1	C:LYS39	NZ	2.7
A:LYS53	NZ	C:GLU97	OE2	3.2
A:LYS53	NZ	C:GLU104	OE2	2.8
A:HIS56	NE2	C:GLU106	OE2	3.3
A:ARG69	NH2	C:GLU35	OE1	3.1
A:ARG69	NH2	C:GLU35	OE2	2.9
A:ARG69	NH1	C:GLU100	OE1	3.2
A:ARG69	NH1	C:GLU35	OE1	2.8
A:LYS96	NZ	C:GLU97	OE1	2.7
B:GLU54	OE2	D:LYS6	NZ	2.6
B:ASP100	OD1	D:LYS39	NZ	2.7
B:LYS53	NZ	D:GLU106	OE1	3.4
B:LYS53	NZ	D:GLU97	OE2	2.7
B:ARG69	NE	D:GLU35	OE1	3
B:ARG69	NH1	D:GLU35	OE1	3.4
B:ARG69	NH1	D:GLU35	OE2	3
B:LYS96	NZ	D:GLU97	OE1	2.5

**Table 3 molecules-27-05574-t003:** Water-mediated hydrogen bonds between IMM01 and CD47.

IMM01		Distance (Å)	Water ID	CD47		Distance (Å)
Residue	Atom			Residue	Atom	
A:ASP100	OD2	3.5	24	C:LYS39	NZ	3.4
				C:LYS41	NZ	3.5
				C:GLU97	OE2	2.9
A:LYS96	NZ	2.8	58	C:THR99	OG1	2.5
				C:GLU104	OE2	2.7
				C:GLU97	OE1	3.5
A:ILE36	O	2.7	57	C:GLU104	OE1	2.8
A:GLN52	N	2.9				
A:GLU54	N	3	55	C:GLU106	N	3
				C:GLU106	O	3
	OE1	3.2	47	C:PHE4	N	2.9
A:ARG69	O	3.3	93	C:THR102	N	3.4
				C:ARG103	N	3.5
B:ASP100	OD1	3.4	5	D:LYS39	NZ	3.3
				D:GLU97	OE2	3.1
B:LYS96	NZ	2.9	41	D:THR99	OG1	2.7
				D:GLU104	OE2	2.8
B:GLU54	N	3.1	36	D:GLU106	O	2.9
					N	3
B:ARG69	N	3.1	90	D:GLU29	OE1	2.6
B:GLU70	N	3.1			OE2	3.5
B:ARG69	NH2	3	16	D:GLU29	OE1	3.5

## Data Availability

The datasets analyzed during the current study are not publicly available.

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
