# Peer review of "Crystal Structure of Human CD47 in Complex with Engineered SIRPα.D1(N80A)"

_molecules, 2022, doi:10.3390/molecules27175574_

Round 1

Reviewer 1 Report

In this study, the authors have released an IMM01/CD47 crystal structure elaborating the potential clinical applications of IMM01. The paper is technically good. As a validation, the released structure has a high similarity to the reported CD47/ SIRP alpha complex. The paper is acceptable with minor English proofreading. For example, 

- Line 62, (including a NH2-terminal ligand binding), should be (an NH2-terminal ligand binding).

This statement need revision

Line 352, (Our data on 353

(the interface structure analysis of the complex of CD47 and IMM01 demonstrated the similarity and difference of the common hydrogen bonds, the salt bridge interaction, and the water-mediated hydrogen bond between IMM01 and CD47. Furthermore, the com-)

Author Response

In this study, the authors have released an IMM01/CD47 crystal structure elaborating the potential clinical applications of IMM01. The paper is technically good. As a validation, the released structure has a high similarity to the reported CD47/ SIRP alpha complex. The paper is acceptable with minor English proofreading. For example, 

- Line 62, (including a NH2-terminal ligand binding), should be (an NH2-terminal ligand binding).

Authors’ response: Thanks for your suggestion. Manuscript has been revised:

This statement need revision

Line 352, (Our data on 353 (the interface structure analysis of the complex of CD47 and IMM01 demonstrated the similarity and difference of the common hydrogen bonds, the salt bridge interaction, and the water-mediated hydrogen bond between IMM01 and CD47. Furthermore, the com-)

Authors’ response: Thanks for your suggestion. Manuscript has been revised.

Our data demonstrated the crystal structures of the complex of IMM01 and CD47 on the common hydrogen bonds, the salt bridge interaction, and the water-mediated hydrogen bonds.

Reviewer 2 Report

The manuscript reports the structural analysis of the complex of CD47 and IMM01, a novel SIRPα-Fc fusion protein which relates which is overexpressed on many types of tumors and acts as an important tumor antigen. The manuscript lies in the journal scope and can be accepted after taking into account a number of comments and corrections: 

- In the structured abstract, the background paragraph needs to be reformulated as the presented sentence just mentions the aim. It lacks the expression role in tumor enhancement and background information

- Page1, line 19-21, please, rewrite in a more correct form, suggestion is: add "showed that" before "the interaction"

- Page 1, line 23, please add "in" or "For" at the beginning of the sentence "The structure of human....."

- I suggest the introduction part is preferred to be provided a paragraph related to a background referring to structural investigation on CD47 with leading citing reference(s).

- In the materials and methods part: The Construction, expression, and purification of CD47-His part did not contain any references. Is it a new procedure,  If so, please, mention that information.

- Page 2, lines 85-88, starting from "The plasmid was.................... needs to be rewritten in a more clear form

- Page, 11, line 289, please rewrite in a more clear and correct form; suggestion:  add "were" 

- Page, 13, lines 364-365; rewrite in a more correct form.

- Conclusion part; is brief and requires to provide more about the afforded results 

- The reference part referring to the introduction part, need to be increased

Author Response

Comments and Suggestions for Authors

The manuscript reports the structural analysis of the complex of CD47 and IMM01, a novel SIRPα-Fc fusion protein which relates which is overexpressed on many types of tumors and acts as an important tumor antigen. The manuscript lies in the journal scope and can be accepted after taking into account a number of comments and corrections: 

- In the structured abstract, the background paragraph needs to be reformulated as the presented sentence just mentions the aim. It lacks the expression role in tumor enhancement and background information

Authors’ response: Thanks for your great suggestion. Based on your suggestion, the abstract has been revised according to your suggestion and the guideline of the journal.

Targeting CD47/SIRPa signaling pathway represents a novel approach to enhance anti-tumor immunity. However, the crystal structure of the CD47/SIRPa has not been fully studied. This study aims to analyze the structure interface of the complex of CD47 and IMM01, a novel recombinant SIRPα-Fc fusion protein.

- Page1, line 19-21, please, rewrite in a more correct form, suggestion is: add "showed that" before "the interaction"

Authors’ response: Revision has been made:

Comparison with the literature structure (PDB ID 2JJT), ) showed that the interactions of IMM01/CD47 structure are the same.

- Page 1, line 23, please add "in" or "For" at the beginning of the sentence "The structure of human....."

Authors’ response: Revision has been made:

 In the structure of human CD47 in complex with human SIRPα, except SER66, the amino acids that ……….

- I suggest the introduction part is preferred to be provided a paragraph related to a background referring to structural investigation on CD47 with leading citing reference(s).

Authors’ response: Thanks for your suggestion. Manuscript has been revised:

Regarding the crystal structures of CD47/SIRPα complex, Hatherley et al described the high-resolution X-ray crystallographic structures of the immunoglobulin superfamily domain of CD47 alone and in complex with the N-terminal ligand-binding domain of signal regulatory protein alpha (SIRPα). This explained the specificity of CD47 for the SIRP family of paired receptors in atomic detail [15]. However, few studies have been done regarding the crystal structure of the CD47/SIRPα-Fc fusion protein complex. In this study, we explore the crystal structure of the CD47/IMM01, a novel SIRPα-Fc fusion protein,

- In the materials and methods part: The Construction, expression, and purification of CD47-His part did not contain any references. Is it a new procedure,  If so, please, mention that information.

Authors’ response: Thanks for your questions. The preparation and purification of CD47-His protein is a routine experimental operation, and different laboratories may have its own set of operating procedures. Three references regarding protein expression and his tag protein purification have been added.

- Page 2, lines 85-88, starting from "The plasmid was.................... needs to be rewritten in a more clear form

Authors’ response: Revision has been made:

The plasmid was extracted and transiently transfected into EXPI 293f cells to express the target protein.

- Page, 11, line 289, please rewrite in a more clear and correct form; suggestion:  add "were" 

Authors’ response: Thanks for your suggestion. Revision has been made:

In the Protein Data Bank, 2JJT was designated as the identifier for the CD47/ SIRPa complex (PDB ID: 2JJT)

- Page, 13, lines 364-365; rewrite in a more correct form.

Authors’ response: Thanks for your suggestion. Revision has been made:

As a new SIRPa-Fc fusion protein targeting the CD47/SIRPa pathway, IMM01 exhibited strong dual-functional anti-tumor activity through phagocytosis by blocking the "don't eat me" signal and activating the "eat me" signal. IMM01 can be used as a monotherapy or in combination with other targeted immune checkpoint inhibitors.

- Conclusion part; is brief and requires to provide more about the afforded results 

Authors’ response: Thanks for your suggestion. Revision has been made by adding more results.

Separate results to be published show that IMM01 has strong dual-functional anti-tumor activity through phagocytosis and removal of N-glycosylation modification improves the potency.

- The reference part referring to the introduction part, need to be increased

Authors’ response: Thanks for your suggestion. Revision has been made by adding more content and references in the introduction part and 6 references have been added.